# Preparation, Properties and Therapeutic Effect of a TPL Nanoparticle Thermosensitive Gel for Intra-Articular Injection

**DOI:** 10.3390/molecules28124659

**Published:** 2023-06-09

**Authors:** Lijuan Wang, Yongliang Ding, Qian Tang, Xiaodong Niu

**Affiliations:** Department of Pharmacy, Chongqing Engineering Research Center of Pharmaceutical Sciences, Chongqing Medical and Pharmaceutical College, Chongqing 401331, China

**Keywords:** triptolide, thermosensitive gel, nanoparticle, intra-articular injection, therapeutic effect

## Abstract

Most injectable preparations for the articular cavity are solution-type preparations that are frequently administered because of rapid elimination. In this study, triptolide (TPL), an effective ingredient in the treatment of rheumatoid arthritis (RA), was prepared in the form of a nanoparticle thermosensitive gel (TPL-NS-Gel). The particle size distribution and gel structure were investigated by TEM, laser particle size analysis and laser capture microdissection. The effect of the nanoparticle carrier material PLGA on the phase transition temperature was investigated by ^1^H variable temperature NMR and DSC. The tissue distribution, pharmacokinetic behavior, four inflammatory factors and therapeutic effect were determined in a rat RA model. The results suggested that PLGA increased the gel phase transition temperature. The drug concentration of the TPL-NS-Gel group in joint tissues was higher than that in other tissues at different time points, and the retention time was longer than that of the TPL-NS group. After 24 days of administration, TPL-NS-Gel significantly improved the joint swelling and stiffness of the rat models, and the improvement degree was better than that of the TPL-NS group. TPL-NS-Gel significantly decreased the levels of hs-CRP, IL-1, IL-6 and TNF-α in serum and joint fluid. There was a significant difference between the TPL-NS-Gel and TPL-NS groups on Day 24 (*p* < 0.05). Pathological section results showed that inflammatory cell infiltration was lower in the TPL-NS-Gel group, and no other obvious histological changes were observed. Upon articular injection, the TPL-NS-Gel prolonged drug release, reduced the drug concentration outside the articular tissue and improved the therapeutic effect in a rat RA model. The TPL-NS-Gel can be used as a new type of sustained-release preparation for articular injection.

## 1. Introduction

Intra-articular injection is a local drug delivery method that releases drugs directly into the diseased joint, which can improve the concentration of drugs in the articular cavity and reduce systemic adverse drug reactions [1,2]. It is often used in the treatment of osteoarthritis (OA) [3,4] and rheumatoid arthritis (RA) [5,6,7]. However, most of the intra-articular injection preparations used in clinical practice are administered in solution form, and the drug is quickly removed from the joint cavity and penetrates into the systemic circulation after administration [8,9,10]. This directly leads to short retention time and the frequent administration of drugs in the joint cavity, which reduces medication compliance. At present, some novel drug delivery systems are used in articular cavity injection research, such as liposomes, microspheres, nanoparticles, nanomicelles, and thermosensitive gels [11,12,13,14,15]. However, the single drug delivery system is difficult to apply in clinical practice due to the limitation of the drug loading dose, obvious sudden release effect and needle transmissibility. The different methods of codelivery may be used in combination for better drug delivery [16,17,18]. In this paper, the PLGA nanoparticles and a poloxamer thermosensitive gel were combined to overcome the shortcomings of drug loading, sudden release and needle connectivity.

PLGA is a good biocompatible, biodegradable material. Its degradation product in the body, lactic acid monomers, can be used as in the Krebs cycle, and it does not cause any obvious inflammation, immune response or cell toxicity. PLGA can be used for a long time without accumulation in the body, delaying the drug release time, improving the drug half-life, and reducing drug toxicity. Poloxamer 407 (P407) is a three-stage copolymer (PEO-PPO-PEO) composed of polyoxyethylene (PEO) and polypropylene (PPO), and its aqueous solution has temperature-sensitive properties [19]. Compared with natural temperature-sensitive polymer materials, P407 can be used to predesign the proportion of the three-stage copolymer and control the physical and chemical properties, mechanical strength and degradation time of the gel. Compared with other synthetic temperature-sensitive materials, it has the advantages of low toxicity, bioinertness and good biocompatibility. When P407 is administered via injection, it has the advantages of minimal trauma, delayed drug release, reduced administration times, low toxicity and immunogenicity. Poloxamer thermosensitive gels are widely used in drug delivery systems, such as nasal, eye, vaginal, periodontal, skin, tumoral, joint and other tissues [20,21,22,23], due to their rapid gelation, suitable viscoelasticity and special three-dimensional mesh structure. 

In this study, triptolide (TPL), an effective ingredient in the treatment of RA, was wrapped into nanoparticles with PLGA as the carrier material (TPL-NS). TPL-NS was dispersed into a solution of P407 at a certain concentration to form a P407 thermosensitive gel for joint cavity injection (TPL-NS-Gel). An effective therapeutic concentration was achieved by adjusting the amount of nanoparticles; the sudden release effect of nanoparticles was controlled by transforming the thermosensitive gel into a gel with high viscosity at body temperature, and the needle connectivity was improved by controlling the size of the nanoparticles and the fluidity of the thermosensitive gel at low temperature. The properties of TPL-NS-Gel were investigated, and its therapeutic effect was evaluated in a rat model of RA.

## 2. Results

### 2.1. Characterization of TPL-NS-Gel

Freeze-dried TPL-NS were white loose powder. When the microspheres were dispersed in water, they were in the state of a uniform emulsion with blue opalescence (Figure 1A). The particle size of the microspheres measured by TEM was approximately 100 nm (Figure 1B). The size distribution of the microspheres measured by a Malvern laser particle size analyzer showed that the particle size was fairly uniform (Figure 1C). The encapsulation efficiency (EE%) and loading efficiency measured by the UPLC-MS method was approximately 83.55 ± 2.67% and 4.73 ± 1.05%, respectively. The poloxamer gel was a flowing liquid at a low temperature (below 20 °C) and formed a soft semisolid gel at 37 °C (Figure 1D). The internal structures of TPL-NS-Gel at 4 °C and 35 °C were detected with a laser capture microdissection system, which showed a radiation micellar shape of more than 30 μm at 35 °C (Figure 1E).

### 2.2. Effect of PLGA on the Phase Transition Temperature of Poloxamer Gel

The hydrogen NMR spectra of poloxamer thermosensitive gel (298 K, Figure 2A) showed that the chemical shifts of EO-CH_2_-, PO-CH_2_- and PO-CH_3_ were approximately 3.6, 3.4–3.6 and 1.1 ppm, respectively. The protons on poloxamer molecules all moved to the low field (high ppm) with increasing temperature (28–32 °C); that is, the protons moved in the direction of decreasing polarity, indicating that the chemical environment of protons changed from hydrophilic to hydrophobic, and the hydrophobic effect between poloxamer molecules was enhanced. However, the chemical shifts of different protons were different (Figure 2B), indicating that the sensitivity to temperature was different. The chemical shift of protons on PO-CH_2_- can reach 1.23 × 10^−2^ ppm/°C, and the chemical shift of protons on PO-CH_3_ can reach 1.04 × 10^−2^ ppm/°C. TPL-NS-Gel containing 1%, 5% and 10% TPL was placed at different temperatures (18 °C, 20 °C, 22 °C, 24 °C, 26 °C, 28 °C, 30 °C and 32 °C) to monitor the chemical shift of protons (Figure 2C). The results show that the chemical shift of the proton on PO-CH_2_- and PO-CH_3_ increases in temperature before the gelling temperature, and the chemical shift of the proton on PO-CH_2_- and PO-CH_3_ changes obviously near the phase transition temperature. Moreover, with increasing TPL-NS content, the phase transition temperature of the P407 gel increased. The phase transition temperatures of 1%, 5% and 10% TPL-NS-Gel were 24–26 °C, 28–30 °C and 30–32 °C, respectively, which were consistent with the gelling temperatures measured by the magnetic stirring method (24.1 °C, 28.8 °C and 32.3 °C).

DSC diagram of poloxamer gel showed that there was an obvious endothermic peak in the formation process of gel. When no PLGA nanoparticles were added, the endothermic peak appeared at 27.57 °C (Figure 3A), and when 5% PLGA nanoparticles were added, the endothermic peak appeared at 29.84 °C (Figure 3B). PLGA nanoparticles increased the endothermic peak temperature, which was close to the gelation temperature measured by the magnetic stirring method (28.8 °C). 

### 2.3. Changes of Joint Status before and after Administration in Rat Model of RA

Redness and swelling began to appear in SD rats 3–5 days after injection of chicken type II collagen and Friedrin’s complete adjuvant, and the swelling became increasingly obvious after that. Generally, the swelling and joint stiffness reached a very obvious degree in approximately two weeks, accompanied by a certain reduction in diet and activity, indicating that the rat RA model was successfully constructed. After 24 days of administration, the photos of diseased joint sites of experimental animals in the TPL-NS-Gel group, TPL-NS group and control group were observed (Figure 4). The results showed that the degree of joint swelling, bending and stiffness of rats in the control group were aggravated with the extension of time, and the symptoms of incomplete feeding and reduced activity appeared. In the TPL-NS-Gel group, the degree of joint swelling and bending stiffness were significantly improved, and there was no significant decrease in the amount of eating and activity and no weight loss. In the TPL-NS group, the redness and swelling of the joints were alleviated to a certain extent, but the stiffness was not significantly improved. In the control group, the degree of swelling and stiffness was further developed, especially the degree of swelling, which was significantly aggravated. 

### 2.4. Changes of Inflammatory Cytokines in Rat Model of RA before and after Administration

Figure 5 shows the standard curves and regression equations of ELISA methods for the determination of four inflammatory factors, IL-1, IL-6, TNF-α and hs-CRP, with R^2^ values above 0.994, meeting the methodological requirements. Approximately 4–5 weeks after the establishment of the RA model, the levels of four inflammatory factors in the articular fluid and serum of rats were significantly increased compared with those of healthy rats (*p* < 0.01). Table 1 shows the content determination results of four inflammatory factors in serum and joint fluid of the TPL-NS-Gel group, TPL-NS group and control group before and after administration. The data showed that the contents of hs-CRP, IL-1, IL-6 and TNF-α in serum and articular fluid of the rat RA model significantly decreased in the TPL-NS-Gel group and TPL-NS group compared with the control group after 12 days of administration (*p* < 0.01). There was no significant difference between the TPL-NS-Gel group and TPL-NS group in other inflammatory factors except TNF-α. After 24 days of administration, the contents of hs-CRP, IL-1, IL-6 and TNF-α in serum and joint fluid of the rat RA model were still significantly decreased in the TPL-NS-Gel group and TPL-NS group compared with the control group (*p* < 0.01). The four inflammatory factors were also significantly different between the TPL-NS-Gel group and the TPL-NS group (*p* < 0.05) which indicated that the long-term therapeutic effect of the TPL-NS-Gel group was better than that of the TPL-NS group. On Day 24, compared with the control group, the four inflammatory factors of the TPL-NS-Gel group in serum decreased by 73.79%, 42.00%, 66.32% and 67.99%, and the factors in joint fluid decreased by 74.29%, 42.27%, 73.55% and 82.46%, respectively. Compared with the control group, the percentage of factors in serum decreased by 60.78%, 27.48%, 59.51% and 58.55%, and the percentage of factors in joint fluid decreased by 61.97%, 29.33%, 69.72% and 69.57%, respectively. However, the levels of four inflammatory factors in the control group were continuously increased. At 24 days after administration, there were significant differences in the other three indicators (*p* < 0.05), except hs-CRP in serum was not significantly increased (*p* = 0.19), indicating that the degree of inflammatory factors in the RA model without treatment increased. For the same indicators in the same group, the decrease percentage in serum was smaller than that in joint fluid, which may be due to the higher expression of these indicators in RA joints than in serum, while TPL exerts its anti-RA effect mainly by reducing the levels of inflammatory factors such as hs-CRP, IL-1, IL-6, and TNF-α.

### 2.5. Results of UPLC-MS/MS Methodology Investigation

The secondary mass spectra of tissues and plasma of rats showed that the endogenous substances did not interfere with the determination. The degree of separation between TPL (retention time was approximately 3.1 min) and internal standard (retention time was approximately 2.4 min) was above 1.5. The detection limit was 0.2 ng/mL, and the quantification limit was 2 ng/mL. The standard curve and correlation coefficient of the heart, liver, spleen, lung, kidney, plasma and joint tissue are shown below. y = 0.0306x + 0.1622 (R^2^ = 0.9977), y = 0.0303x + 0.1284 (R^2^ = 0.9958), y = 0.0301x + 0.1452 (R^2^ = 0.9974), y = 0.0316x + 0.0773 (R^2^ = 0.9936), y = 0.0323x + 0.1217 (R^2^ = 0.9962), y = 0.0316x + 0.0825 (R^2^ = 0.9938), y = 0.0254x + 0.1231 (R^2^ = 0.9979). The linear ranges were 2.47–247.25 ng/g (ng/mL), and the relative recoveries were ≥88.3%. The RSDs of intra-day and inter-day precision were ≤4.37% and 6.71%, respectively. The RSDs of stability were ≤4.46% at room temperature for 8h and ≤3.85% after three repeated freeze-thaw cycles. The matrix effect was ≥92.8%.

### 2.6. The Distribution of Tissues and Plasma Pharmacokinetic Behavior in RA Rats

The tissue distribution data after the injection of TPL-NS-Gel and TPL-NS into the articular cavity are shown in Table 2. The results showed that the peak concentration in the articular tissue of the TPL-NS-Gel group was 102.64 ± 9.47 ng/g, the peak time was the 8th day, and the articular cavity retention time was more than 24 days. The peak concentration in the joint tissue of the TPL-NS group was 197.18 ± 9.54 ng/g, the peak time was the third day, and the retention time of the joint cavity was approximately 2 weeks. Compared with the TPL-NS group, the peak concentration of TPL-NS-Gel in the joint was lower, but the retention time was longer, indicating that the thermosensitive gel could further delay drug release, reduce the rapid elimination of solution form in the joint cavity, and further reduce the number of drug administration in the joint cavity. After administration in the two groups, the drug concentration in other tissues (heart, liver, spleen, lung, kidney, and plasma) except joints was relatively low, suggesting that local drug administration in the joint cavity itself had a certain local targeting effect. However, the peak concentration in the TPL-NS-Gel group was significantly lower than that in the TPL-NS group, indicating that drug accumulation in the joint cavity was beneficial to reduce drug concentration in all tissues, reducing systemic toxicity and side effects, and facilitating the advantages of local drug treatment. The pharmacokinetic parameters were calculated using a non-compartment model, and the main pharmacokinetic parameters are listed in Table 3. The AUC_(0–-t)_ and MRT_(0–t)_ of the TPL-NS-Gel group were 1.7 and 2.3 times that of the TPL-NS group, respectively. Based on the concentration–time curves (Figure 6), the in vivo pharmacokinetic data of TPL-NS-Gel and TPL-NS on RA rats were also fitted to a two-compartment model, and the main pharmacokinetic parameters are listed in Table 3. The AUC_(0–t)_ of the TPL-NS-Gel group was 1.7 times that of the TPL-NS group, which did not show an obvious difference from that of the non-compartment model. The t_1/2_Ka, t_1/2_α and t_1/2_β of the TPL-NS-Gel group were 8.1, 2.4 and 1.6 times that of the TPL-NS group, respectively. It suggested that the poloxamer gel formed an obvious drug reservoir in the joint cavity, prolonging the absorption, distribution and elimination of the drug in vivo.

### 2.7. Pathological Changes in the RA Rat Model

The heart, liver, spleen, lung, kidney and joint tissues of rats in the TPL-NS-Gel group, TPL-NS group and control group were taken for HE staining 24 days after administration. The pathological morphology was observed as shown in Figure 7. HE staining of joint tissue showed that the inflammatory cell infiltration, the degree of tissue hyperplasia, and the degree of cell swelling and deformation were significantly reduced compared with the control group, and the number and dilation of blood vessels were alleviated in the TPL-NS-Gel Group. In the TPL-NS group, other phenomena were not significantly improved except for the decrease in inflammatory cell infiltration. HE staining of heart, liver, spleen, lung and kidney tissues showed no significant difference in TPL-NS-Gel Group and TPL-NS Group compared with the control group, no obvious histological changes, inflammatory cell infiltration, etc. The HE results showed that local injection of TPL-NS-Gel in the joint cavity had a good effect on the improvement of joint histopathologic morphology, and no systemic organ toxicity was observed.

## 3. Discussion

In recent years, thermosensitive gels have been commonly used to treat joint diseases [24,25]. The poloxamer-based gel was widely used in drug delivery systems because of its special advantage. There have been two main approaches to studying poloxamer thermosensitive gels. One is to build a gel delivery system together with carbomer, hyaluronic acid, hydroxypropyl methyl cellulose and other biological adhesive materials to solve the problem of poor biological adhesion. The second is to combine poloxamer thermosensitive gels with inclusion compounds, microspheres, liposomes, nanoparticles, micelles and other new technologies to improve drug solubility, enhance drug stability, reduce systemic side effects, prolong drug release time and so on [16]. In this study, PLGA nanoparticles were combined with a poloxamer thermosensitive gel using TPL as the model drug, and hyaluronic acid was added to improve gel adhesion. The prepared TPL-NS-Gel had the following characteristics. First, the small particle size of the nanoparticles (less than 200 nm) and the fluidity of the gel at low temperatures ensured good needle connectivity. Second, the drug loading capacity of nanoparticles is small, and the use of high doses will bring potential risks to the drug site. The slow release after the formation of the gel is conducive to reducing the high load of local drugs. Third, the dual slow-release characteristics of nanoparticles and thermosensitive gels increase the retention time in the joint cavity.

The ultrasonic velocity, dynamic scattering, light scattering, small-angle X-ray diffraction, small-angle neutron scattering, rheology and electrolyte behavior of the P407 aqueous solution were studied [26,27,28,29]. It was found that the gelation process was the result of micellar properties (aggregation number and micellar symmetry, etc.) changes, micellar polymerization and interaction [30]. The hydrophobic polypropylene oxide groups in poloxamer micelles dehydrate and swell after temperature rise. When the temperature reaches a certain level, the micelles begin to contact with each other and form a gel with a network structure. The ^1^H NMR results showed that the chemical shifts of different protons on poloxamer molecules moved to the lower field obviously near the phase transition temperature. This means that there is a discontinuous change in the chemical environment of the proton during the transition from solution to gel. At low temperatures, the solution of P407 was in a free-flowing state. Before the phase transition temperature, poloxamer molecules gradually self-polymerized to form micelles with increasing temperature. When it reached the vicinity of the phase transition temperature, the micelles arranged regularly, and the hydrophobic force between Poloxamer molecules increased, forming a three-dimensional network structure. The temperature corresponding to the turning point of the chemical shift of protons on PO-CH_2_- and PO-CH_3_ (28–29 °C) is close to the gelling temperature measured by the magnetic stirring method (30.3 °C). Since PLGA is the carrier material of nanoparticles, it is necessary to study the effect of PLGA on the temperature of poloxamer gel. The ^1^H NMR results showed that the change of poloxamer solution to gel was closely related to the change of micellar properties. The addition of PLGA affected the chemical shift of protons on PO-CH_2_- and PO-CH_3_ and increased the gel phase transition temperature. As more PLGA nanoparticles were added, the phase transition temperature continued to increase. DSC was one of the common methods to study the thermal gelation kinetics, which reflected the thermal change of solution during the transition to gel [31]. After the addition of PLGA nanoparticles, the poloxamer system micellized and gelled at a higher temperature, which was consistent with the results of ^1^H variable temperature NMR. It is speculated that there are two possible reasons. First, PLGA nanoparticles are dispersed into nano-suspension in water, and the particle structure enters poloxamer micelles, affecting the formation of micelles. At the same time, it can support the micellar, which is conducive to increasing the gel strength and prolonging the release of drugs. Second, the carboxyl group of some PLGA interacts with the poloxamer ether bond, which interferes with the arrangement of micelles into a three-dimensional network structure. Only when the temperature increased further could the micelles be dehydrated and polymerized into a gel.

Indices to evaluate the efficacy of joint injectable preparations in the treatment of RA include joint swelling stiffness, inflammatory factor content, tissue distribution concentration, blood drug concentration, pathological sections, etc. In this paper, the changes in the above indices in the TPL-NS-Gel group and TPL-NS group after 24 days of administration were recorded. RA is a chronic systemic autoimmune disease with high morbidity, high disability rate, and a serious impact on patient quality of life. It mainly manifests as severe limb deformity and widespread inflammation caused by joint lesions. In the course of RA, repeated swelling and effusion of one or several joints often occur. Improving the degree of joint swelling and stiffness has great significance for restoring normal movement in patients. From the joint appearance photos before and after administration, it could be seen that the TPL-NS-Gel group had a better effect on improving joint stiffness and swelling than the TPL-NS group. Inflammatory cytokines are the main pathologic mediators of rheumatoid arthritis. IL-1 and IL-6, which are mainly secreted by mononuclear macrophages, are the most important inflammatory factors in the destruction of articular cartilage. TNF-α mediates the infiltration of arthritic cells by inducing endothelial cell adhesion molecules and acts as an activating factor of osteoclasts to cause the absorption and destruction of bone and cartilage, promoting fibroblast proliferation. Hs-CRP is a nonspecific immune response indicator, and its level is closely related to changes in the disease, which is of great value for the treatment and evaluation of RA. TPL is an inflammatory inhibitor whose anti-inflammatory activity has been well documented. Its anti-inflammatory effect is realized by regulating inflammatory cytokines, inflammatory mediators and nuclear transcription factors. Therefore, the activity of TPL can be evaluated by detecting the changes in inflammatory cytokines before and after administration. The levels of four inflammatory factors before and after administration demonstrated that the long-term sustained release effect of the TPL-NS-Gel group was significantly better than that of the TPL-NS group at the same dosage. The tissue distribution and blood concentration showed different distribution behaviors after injection of TPL-NS-Gel and TPL-NS into the joint cavity. Drug reservoirs were formed in the joint cavity in both groups, gradually releasing drugs, but the peak concentration in the TPL-NS-Gel group was approximately half of that in the TPL-NS group, the peak time was extended to 1.6 times, and the joint cavity retention time was as long as 3–4 weeks. Before reaching the peak concentration, the concentration of the TPL-NS-Gel group in each tissue was lower than that of the TPL-NS group, indicating that the thermosensitive gel prevented the sudden release effect of the nanoparticle solution to a certain extent. After reaching the peak concentration, the concentration of the TPL-NS-gel group in each tissue was higher than that of the TPL-NS group, and the effect of the thermosensitive gel on further delaying drug release was verified. In summary, after the nanoparticles were made into a thermosensitive gel, the retention time in the joint cavity was longer, the sudden release effect was weakened, and the sustained release effect of the drug was more significant.

The drug concentrations in the heart, liver, spleen, lung, kidney and plasma in the TPL-NS-gel and TPL-NS groups were significantly lower than those in the joint tissues, indicating that drug administration in the joint cavity could reduce the accumulation of drugs in non-target organs and enhance the advantages of local drug treatment. However, the peak concentration in the TPL-NS-Gel group was significantly lower than that in the TPL-NS group. These results indicated that the drug accumulation effect of the thermosensitive gel was more conducive to reducing drug concentration in various tissues and alleviating systemic toxicity and side effects. After the end of the administration, joint tissues were taken for HE pathological sections, and the results of HE staining indicated the improvement of joint histopathologic morphology from the microscopic perspective.

## 4. Materials and Methods

### 4.1. Materials and Reagents

Standards of TPL (the purity is above 98.5% determined by HPLC-UV, China National Institutes for Food and Drug Control); PLGA (molecular weight = 15,000 Da, Shandong institute of medical devices); methanol (Merck Co., Gradient Grade), acetonitrile (Merck Co., Gradient Grade), ammonium formate (Merck Co., Gradient Grade); Rat IL-1α ELISA Kit, Rat IL-6 ELISA Kit, Rat TNF alpha ELISA Kit (Abcam^®^); Rat CRP (C-Reactive Protein) ELISA Kit (Elabscience^®^); anhydrous ethanol, paraformaldehyde, xylene, formaldehyde, hydrochloric acid, neutral balsam, propylene glycol (Sinopharm Chemical Reagent Co., Ltd., analytical grade); Chicken type II collagen (Hefei Bomei biotechnology Co., Ltd.); Phosphate buffer solution (PBS, Life Technologies^®^).

### 4.2. Animals

Male SD rats (weighing 180~220 g) were supplied by the Animal Centre of Chongqing Medical University (Chongqing, China). All the experiments in this study were approved by the Animal Research Ethics Committee of Chongqing Medical and Pharmaceutical College. All the rats were housed with free access to food and water under a standardized light/dark cycle condition (20~22 °C and 45~65% RH).

### 4.3. Preparation and Properties of TPL-NS-Gel

TPL nanoparticles were prepared by an improved self-emulsifying solvent evaporation method. The organic phase consisting of acetone/anhydrous ethanol dissolved with an appropriate amount of PLGA and TPL was added to poloxamer 188 aqueous phase at a certain concentration under agitation, and all the organic solvent and part of the water were evaporated under pressure to obtain a blue opalescent gel suspension. After centrifugation at high speed (12,000 r/min, 10 °C for 3 h), the precipitates were freeze-dried to obtain white nanoparticle powder. One to two drops of TPL nanoparticle suspension were placed on the copper net, and stained with 2% phosphotungstic acid solution, and the morphology of the nanoparticle was observed under a transmission electron microscope. The average particle size and distribution of the nanoparticle suspensions diluted (1000 times) were measured on the Marvin laser particle size detector. An appropriate amount of nanoparticle was taken and dissolved with dichloromethane, filtered, nitrogen blow-dried, mobile phase dissolved, and filtered through a 0.45 μm filter membrane. The mobile phase volume was adjusted to scale, and mass spectrometry was used to determine the concentration, encapsulation rate and drug-loading ability of the nanoparticles. The thermosensitive gel was prepared by the cold method. P407/P188 (19.5%/5%, *w/w*) was dissolved in pure water, and sodium hyaluronate (0.5%, *w/w*) was added and placed in the refrigerator at 4 °C for 24 h. Under the condition of mild agitation, nanoparticles (5% according to TPL, *w*/*w*) were added and stored in the refrigerator at 4 °C for later use. The morphology of gel micelles was observed by the ArcturusXT™ Laser Capture Microdissection System at 37 °C.

### 4.4. The Effect of Nanoparticle Carrier Material PLGA on the Phase Transition Temperature of P407 Gel Was Studied by ^1^H Variable Temperature Nuclear Magnetic Resonance and DSC

D_2_O was used as the solvent to prepare the D_2_O hydrogel. The preparation process was the same as above. Ordinary gel and D_2_O hydrogel samples (0.5 mL) were loaded into the nuclear magnetic tube. A 5 mm BBQ probe, Bruker BVT3000 digital temperature controller with the following parameters was used: single pulse, D_2_O lock field, a resonance frequency of 500.13 MHz, a pulse delay of 6 s, 16 sampling times, a 90° pulse width of 13.2 μs, D_2_O gel as the sample, and D_2_O calibration.

The heating range was −40–+50 °C and the heating rate was 5 °C/min. The testing mode was DSC-TG with an empty crucible as the reference. Liquid nitrogen cooling, poloxamer gel without PLGA and poloxamer gel with 5% PLGA nanoparticles as samples.

### 4.5. The Establishment of Rat RA Model

All the animal experiments approved by the medical ethics committee of Chongqing Medical and Pharmaceutical College were conducted in accordance with the relevant guidelines and regulations. The chicken type II collagen was mixed and emulsified with Complete Freund’s adjuvant. The 0.25 mL emulsifier was injected subcutaneously into the foot plantarum of SD rats. Immunity was further enhanced with an injection of 0.1 mL emulsifier 7 d later. The body weight and paw swelling of the rats were observed every day after the injection. The serum and joint fluid were collected and the expression levels of the inflammatory factors were measured.

### 4.6. Instrumentation and Conditions of UPLC-MS/MS for the Determination of Biological Samples

Chromatographic analysis was performed using an Agilent 1290 Infinity II ultrahigh-performance liquid chromatography system (UPLC), including a binary pump, an online vacuum degasser, a surveyor auto-sampling system, a column temperature controller and a diode-array detector. The samples were separated on an Agilent ZORBAX SB C_18_ column (2.1 × 100 mm, 1.8 μm) and eluted with a gradient elution of solvent A (water containing 5 mmol/L ammonium formate) and solvent B (acetonitrile) as follows: 30% B (0–3.7 min), 95% B (3.7–4.0 min), 90% B (4.0–10.0 min), and 30% B (10.1–15 min). The column temperature was set at 40 °C, the flow rate was set at 0.4 mL/min, and the injection volume was set at 2 μL. Mass spectrometric detection was carried out on an Agilent 6460 triple-quadrupole mass spectrometer with positive electrospray ionization (ESI), which was connected to the liquid chromatography system. The MRM mass scan mode was implemented. The precursor ion/product ion were *m*/*z* 378.1/361.0 for TPL and *m*/*z* 260.0/116.2 for the internal standard (IS, propranolol hydrochloride). The MS/MS conditions were optimized as follows: fragmentor, 100 V; capillary voltage, 4000 V; nebulizer gas pressure (N_2_), 35 psi; drying gas flow rate (N_2_), 10 L/min and gas temperature, 350 °C.

### 4.7. Method Validation of UPLC-MS/MS

A series of standard working solutions were obtained by further diluting the stock solution of TPL with methanol for method validation. The calibration standard samples for TPL (actual concentrations were 2.47, 24.72, 49.45, 123.62 and 247.25 ng/mL, respectively) were prepared by mixing TPL standard working solutions and the IS solution of the same volume into blank heart, liver, spleen, lung, kidney, joint and plasma tissues. The calibration curves were constructed as the peak area ratios of TPL to IS against the TPL concentrations. The lower limit of quantification (LLOQ) was determined as the concentration of TPL with a signal-to-noise ratio of 10. To determine intra-day and inter-day precision and accuracy, five replicates of QC samples at low (2.47 ng/mL), medium (49.45 ng/mL) and high (247.25 ng/mL) concentration levels were prepared and analyzed on the same day and on three different days. The extraction recovery was determined by calculating the ratio of QC samples obtained at the three concentrations mentioned above in the blank tissues against those originally spiked in methanol (*n* = 5). The short-term stability was evaluated by determining QC samples at room temperature for 12 h. The freeze-thaw stability was determined through three freeze-thaw cycles on consecutive days. The matrix effect was determined by calculating the ratio of QC samples obtained at low, medium and high concentration levels against those in the mobile phase (*n* = 5). The detection of the matrix effect can exclude the influence of endogenous components in those tissues, resulting in ion suppression of the analyte signal. The results of method validation are expressed as the mean ± RSD value.

### 4.8. Animal Experimental Protocol

A total of 66 RA model rats were weighed and randomly divided into three groups: (A) 30 rats in the TPL-NS-Gel group (TPL-NS-Gel was administered into the joint cavity at 10 mg/kg); (B) 30 rats in the TPL-NS group (TPL-NS was administered into the joint cavity, 10 mg/kg); (C) six rats in the control group (0.9% saline was administered into in the joint cavity). All groups received a single dose. The heart, liver, spleen, lung, kidney, joint and whole blood were taken at 30 min, 8 h, 1 d, 2 d, 3 d, 5 d, 8 d, 12 d, 16 d, and 24 d after administration (three animals were sacrificed at each time point, *n* = 3). Whole blood was centrifuged at 12,000 r/min for 10 min to separate the plasma. The joints of the dosing side were removed, rinsed with PBS7.4, and wiped dry, and the whole cartilage synovial tissues were removed and weighed. Other tissues were washed with PBS 7.4, dried and weighed. Both plasma and tissues were stored at −80 °C in a freezer. After administration, the animals were observed and recorded daily for signs of status, including joint swelling and stiffness and activity, and feeding condition. Before tissue removal on Days 12 and 24, the joint cavity was injected with 0.5 mL of sterile NaCl solution, and an additional 0.5 mL of joint fluid was withdrawn and stored frozen at −20 °C. Blood was taken, and serum was separated and stored frozen at −20 °C. These joint fluids and serum were used to detect the inflammatory cytokines hs-CRP, IL-1, IL-6, and TNF-α. Part of the tissues taken out on the 12th and 24th days were immersed and fixed in a 10% formaldehyde solution for pathological examination.

### 4.9. Treatment of Biological Samples

Tissues (0.2 g) or plasma (200 μL) were thawed at room temperature before analysis. After adding 50 μL of IS solution (100 μg/mL), TPL in tissues and plasma was extracted twice with ethyl acetate (total 3 mL) by vortexing for 5 min at a time. The extracted solution was centrifuged for 10 min at 12,000 r/min. The ethyl acetate layers were removed and evaporated at 40 °C under nitrogen. The residues were redissolved in 200 μL methanol and subjected to UPLC-MS/MS analysis. The TPL concentrations in each tissue (ng/g) and plasma (ng/mL) were calculated by the above calibration curves. If necessary, the sample should be diluted and then re-injected to ensure that the concentration is within the linear range.

### 4.10. Measurement of hs-CRP, IL-1, IL-6 and TNF-α in Serum and Synovial Fluid by ELISA

ELISA was used to measure inflammatory cytokines in the serum and synovial fluid. The assays were performed according to the protocols outlined in the TNF-α, IL-1, IL-6 and hs-CRP ELISA kits. Synovial fluid and serum were incubated at 37 °C for 60 min according to the manufacturer’s protocol: a total of 100 µL synovial fluid or serum was added to the TNF-α, IL-1, IL-6 or hs-CRP antibody-coated plate and incubated at 37 °C for 1 h. After adding the biotin-conjugated detecting TNF-α, IL-1, IL-6 or hs-CRP antibody and incubating for 2 h, streptavidin-HRP was added, and 3,3’-5,5’ tetramethylbenzidine was used for development. The optical density value was measured at 450 nm by Multiskan spectroscopy (Thermo Fisher Scientific, Inc.). Experiments were performed in triplicate.

### 4.11. Statistical Analysis

Descriptive statistics include the mean and standard deviation (SD) and were determined with Microsoft Excel 2013. The plasma pharmacokinetic data were calculated using DAS2.0 software. The statistical analysis was performed with SPSS 23.0 software (SPSS, IBM Inc.). The statistical significance of differences between the TPL-NS-Gel group, TPL-NS group and control groups in four inflammatory factors were calculated using the independent-sample t-test. Differences were considered statistically significant and very significant when *p* values calculated were <0.05 and <0.01, respectively.

## 5. Conclusions

In conclusion, compared with the same dose of TPL-NS, the TPL-NS-Gel group had significantly improved joint swelling and stiffness in the rat RA model and significantly reduced levels of IL-1, IL-6, TNF-α and hs-CRP. The retention time of TPL in joint tissues is longer, the peak concentration is lower, and the release is more stable. The concentration is lower in tissues except the joint and the potential systemic toxicity is reduced. The results showed the feasibility of TPL-NS-Gel to prolong drug release and improve therapeutic effects, which lays the foundation for future clinical research.

## Figures and Tables

**Figure 1 molecules-28-04659-f001:**
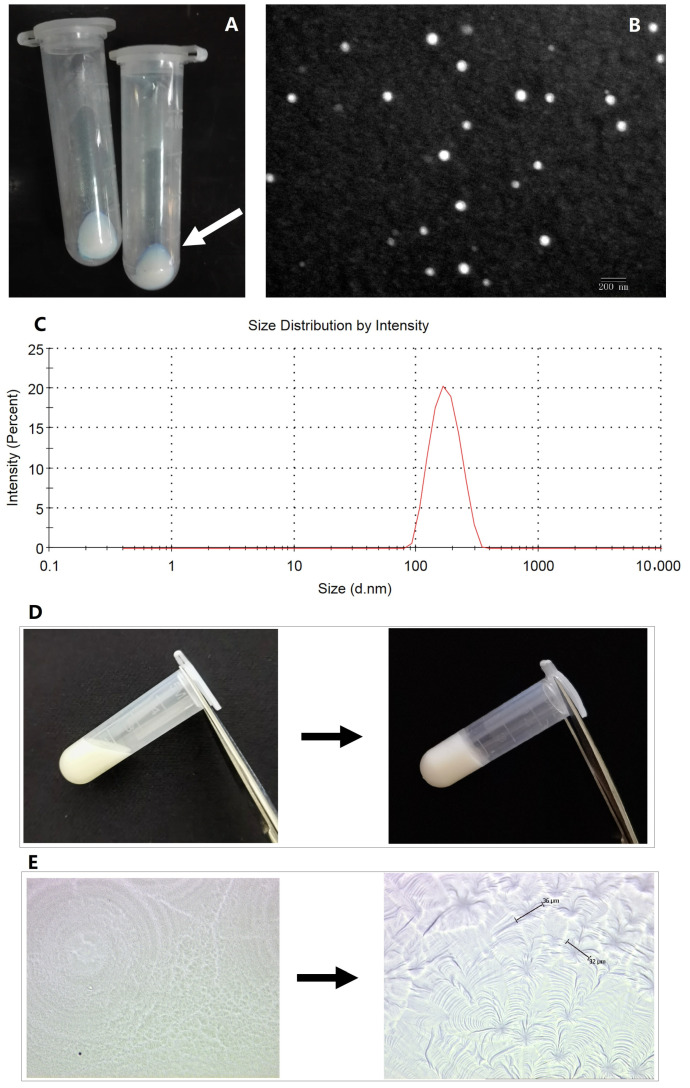
TPL-NS-Gel ((**A**). the uniform emulsion of TPL-NS with blue opalescence as shown by the arrow; (**B**). the particle size of the microspheres measured by TEM; (**C**). The size distribution of the microspheres measured by the Malvern laser particle size analyzer; (**D**). the transition of TPL-NS-Gel from 4 °C to body temperature; (**E**). the internal structures of TPL-NS-Gel from 4 °C to 35 °C were detected with laser capture microdissection system).

**Figure 2 molecules-28-04659-f002:**
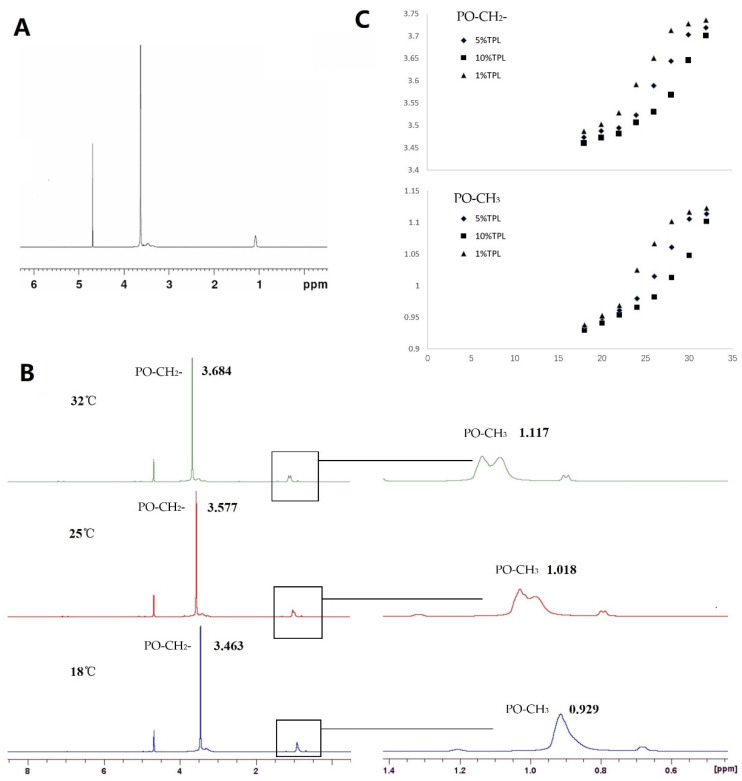
The effect of PLGA on the phase transition temperature of poloxamer gel measured by ^1^H variable temperature NMR. ((**A**). the hydrogen NMR spectra of poloxamer thermosensitive gel; (**B**). the chemical shifts of PO-CH_3_ and PO-CH_2_- at 18 °C, 25 °C, 32 °C; (**C**). the change of chemical shift in PO-CH_3_ and PO-CH_2_- for TPL-NS-Gel containing 1%, 5% and 10% TPL at 18 °C, 20 °C, 22 °C, 24 °C, 26 °C, 28 °C, 30 °C and 32 °C).

**Figure 3 molecules-28-04659-f003:**
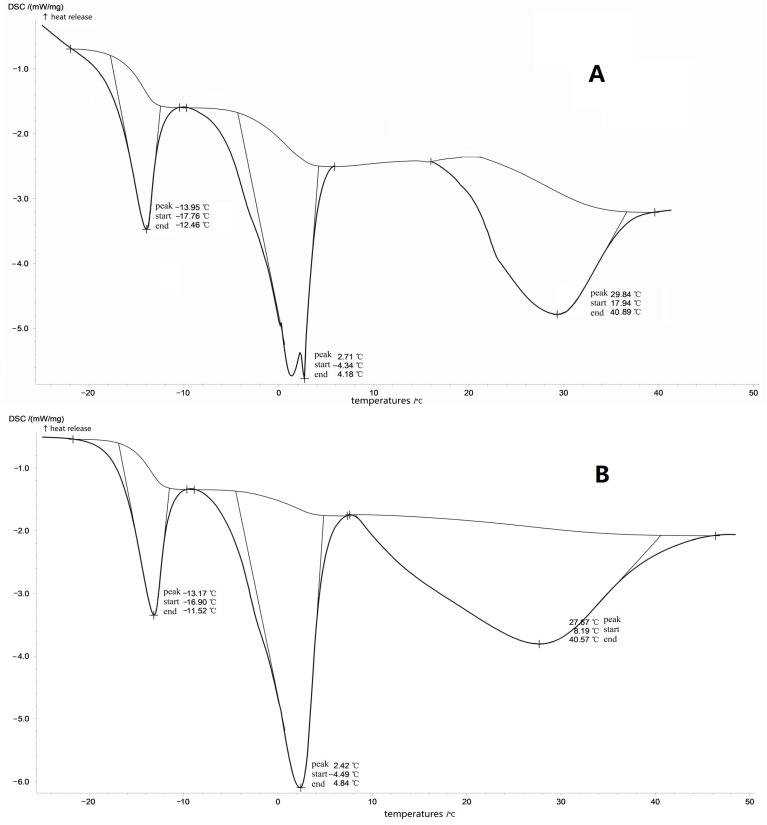
The effect of PLGA on the phase transition temperature of poloxamer gel measured by DSC ((**A**). the poloxamer gel without PLGA nanoparticle; (**B**). the poloxamer gel with 5% PLGA nanoparticle).

**Figure 4 molecules-28-04659-f004:**
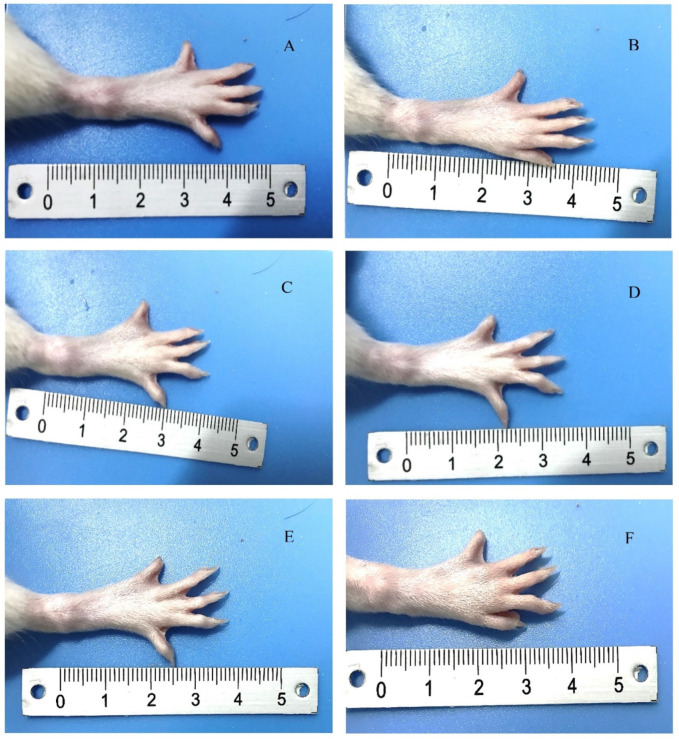
The photographs of the diseased joint in three groups after administration. (**A**). TPL-NS-Gel group (before administration); (**B**). TPL-NS-Gel group (24 days after administration); (**C**). TPL-NS group (before administration); (**D**). TPL-NS group (24 days after administration); (**E**). Control group (before administration); (**F**). Control group (24 days after administration).

**Figure 5 molecules-28-04659-f005:**
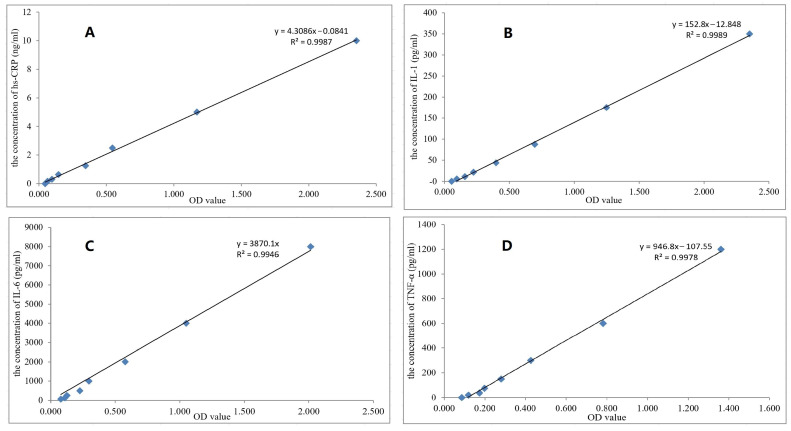
The standard curves and R^2^ of hs-CRP, IL-1, IL-6 and TNF-α in serum and articular fluid were determined by ELISA ((**A**). hs-CRP, (**B**). IL-1, (**C**). IL-6, (**D**). TNF-α).

**Figure 6 molecules-28-04659-f006:**
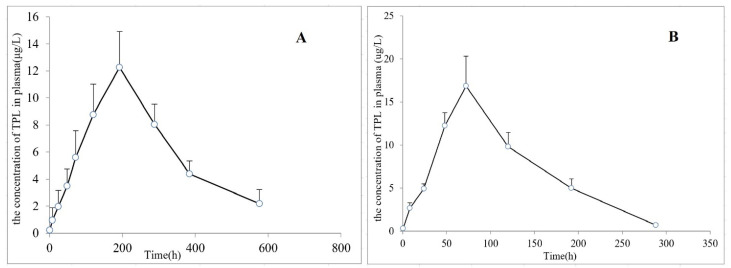
The mean plasma concentration–time curve for TPL-NS-Gel group (**A**) and TPL-NS group (**B**) after articular cavity administration at 10 mg/kg in RA rats (Mean ± SD, *n* = 3). Blood samples were collected at 30 min, 8 h, 1 d, 2 d, 3 d, 5 d, 8 d, 12 d, 16 d, and 24 d after dosing.

**Figure 7 molecules-28-04659-f007:**
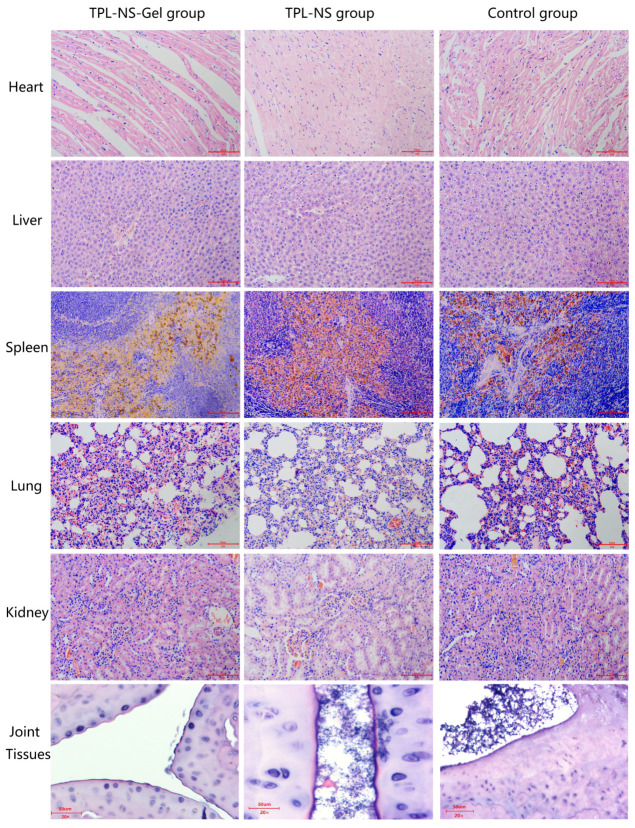
The histological view of different tissues in Rat RA models after administration (HE staining, 20×).

**Table 1 molecules-28-04659-t001:** The concentrations of hs-CRP, IL-1, IL-6, and TNF-α in serum and joint fluid before and after administration and their significance analysis (Mean ± SD, *n* = 3).

Time	Group	Serum				Joint Fluid			
		hs-CRP (ng/mL)	IL-1 (pg/mL)	IL-6 (pg/mL)	TNF-α (pg/mL)	hs-CRP (ng/mL)	IL-1 (pg/mL)	IL-6 (pg/mL)	TNF-α (pg/mL)
before administration	TPL-NS-Gel Group	4.12 ± 0.18	102.48 ± 2.14	616.04 ± 6.40	320.54 ± 2.90	9.57 ± 0.50	154.76 ± 5.81	3022.44 ± 40.53	1187.63 ± 31.46
	TPL-NS Group	4.08 ± 0.17	104.43 ± 2.92	608.22 ± 2.92	320.30 ± 5.25	9.44 ± 0.39	157.53 ± 5.25	3011.61 ± 64.81	1164.06 ± 56.76
	Control Group	3.94 ± 0.57	103.65 ± 2.21	612.01 ± 8.26	317.71 ± 8.58	9.63 ± 0.52	155.33 ± 5.34	2949.85 ± 73.96	1152.28 ± 49.12
	*p* _(TPL-NS-Gel: TPL-NS)_	0.819	0.497	0.191	0.957	0.797	0.644	0.851	0.635
	*p* _(TPL-NS-Gel: Control)_	0.694	0.621	0.614	0.681	0.907	0.924	0.290	0.440
	*p* _(TPL-NS: Control)_	0.763	0.782	0.574	0.734	0.703	0.700	0.425	0.835
12 days after administration	TPL-NS-Gel Group	1.74 ± 0.11	75.77 ± 4.25	280.30 ± 13.46	125.43 ± 5.27	3.50 ± 0.11	112.69 ± 4.16	922.37 ± 24.34	304.96 ± 16.63
	TPL-NS Group	1.87 ± 0.07	81.53 ± 2.91	286.83 ± 9.13	143.67 ± 4.59	3.77 ± 0.21	123.15 ± 6.03	946.02 ± 12.22	373.05 ± 14.06
	Control Group	4.30 ± 0.23	107.69 ± 3.92	646.43 ± 15.59	335.57 ± 5.53	10.69 ± 0.83	166.72 ± 5.70	3073.70 ± 87.89	1232.92 ± 50.41
	*p* _(TPL-NS-Gel: TPL-NS)_	0.232	0.189	0.600	0.021	0.185	0.113	0.287	0.011
	*p* _(TPL-NS-Gel: Control)_	0.000	0.001	0.000	0.000	0.006	0.000	0.000	0.000
	*p* _(TPL-NS: Control)_	0.000	0.002	0.000	0.000	0.000	0.002	0.000	0.000
24 days after administration	TPL-NS-Gel Group	1.08 ± 0.06	59.44 ± 6.54	207.50 ± 10.07	102.61 ± 7.82	2.46 ± 0.12	89.41 ± 2.99	799.37 ± 11.60	208.31 ± 4.90
	TPL-NS Group	1.60 ± 0.13	75.73 ± 3.26	246.29 ± 9.52	132.76 ± 6.47	3.59 ± 0.10	111.33 ± 2.90	912.03 ± 11.89	354.28 ± 10.84
	Control Group	4.60 ± 0.15	118.73 ± 3.56	665.85 ± 21.87	357.96 ± 17.69	13.16 ± 0.62	190.02 ± 2.60	3279.25 ± 31.55	1300.74 ± 17.62
	*p* _(TPL-NS-Gel: TPL-NS)_	0.007	0.034	0.017	0.014	0.000	0.002	0.001	0.000
	*p* _(TPL-NS-Gel: Control)_	0.000	0.000	0.000	0.000	0.000	0.000	0.000	0.000
	*p* _(TPL-NS: Control)_	0.000	0.000	0.000	0.000	0.000	0.000	0.000	0.000

*p* _(TPL-NS-Gel: TPL-NS)_: the significance analysis between TPL-NS-Gel group and TPL-NS group; *p* _(TPL-NS-Gel: Control)_: the significance analysis between TPL-NS-Gel group and control group; *p* _(TPL-NS: Control)_: the significance analysis between TPL-NS group and control group.

**Table 2 molecules-28-04659-t002:** Concentration of TPL determined in tissue following intra-articular administration of TPL-NS-Gel, TPL-NS at 10 mg/kg in RA rats (ng/g or ng/mL, Mean ± SD, *n* = 3).

Group	Time Point	Tissues						
		Heart (ng/g)	Liver (ng/g)	Spleen (ng/g)	Lung (ng/g)	Kidney (ng/g)	Joint (ng/g)	Plasma (ng/mL)
TPL-NS-Gel Group	30 min	0.21 ± 0.09	0.24 ± 0.08	0.29 ± 0.11	0.19 ± 0.07	0.34 ± 0.12	19.52 ± 3.78	0.22± 0.09
	8 h	0.84± 0.27	0.92 ± 0.44	1.15± 0.73	0.62 ± 0.23	1.04 ± 0.51	24.36 ± 3.92	0.95 ± 0.76
	1 d	2.16 ± 0.63	2.73 ± 1.12	3.22 ± 1.38	2.16 ± 0.67	3.26 ± 0.94	45.49 ± 5.16	1.96 ± 0.98
	2 d	3.37 ± 0.91	4.36 ± 0.95	5.06 ± 1.28	3.86 ± 1.03	5.47 ± 1.43	58.27 ± 4.62	3.48 ± 1.05
	3 d	5.03 ± 1.45	7.61 ± 1.39	8.14 ± 1.55	5.88 ± 1.61	7.95 ± 1.68	70.19 ± 5.96	5.59 ± 1.63
	5 d	7.69 ± 2.26	10.72 ± 2.43	11.47 ± 2.23	7.52 ± 1.39	9.39 ± 2.31	89.53 ± 8.82	8.78 ± 1.84
	8 d	10.29 ± 2.37	13.18 ± 2.99	14.09 ± 3.62	9.28 ± 1.64	12.36 ± 2.77	102.64 ± 9.47	12.26 ± 2.17
	12 d	8.32 ± 2.04	11.53 ± 1.76	9.86 ± 1.78	7.11 ± 1.05	9.01 ± 1.45	91.78 ± 9.06	8.05 ± 1.21
	16 d	6.54 ± 1.98	7.80 ± 1.64	5.59 ± 1.16	5.06 ± 0.90	6.93 ± 1.57	73.46 ± 6.68	4.39 ± 0.77
	24 d	3.72 ± 0.83	4.97 ± 1.21	3.44 ± 0.85	2.67 ± 0.79	4.12 ± 0.78	50.89 ± 5.77	2.18 ± 0.85
TPL-NS Group	30 min	0.31 ± 0.08	0.45 ± 0.11	0.39 ± 0.12	0.29 ± 0.13	0.36 ± 0.14	27.61 ± 3.13	0.32± 0.11
	8 h	2.63 ± 0.82	4.16 ± 1.27	4.38 ± 0.71	2.93 ± 0.87	2.29 ± 0.76	63.95 ± 4.37	2.69 ± 0.52
	1 d	5.67 ± 1.16	7.51 ± 1.64	6.65 ± 1.35	6.07 ± 1.26	4.66 ± 1.32	89.36 ± 6.48	4.93 ± 0.47
	2 d	11.35 ± 2.47	10.93 ± 2.72	12.46 ± 1.88	11.27 ± 2.24	10.38 ± 1.54	120.69 ± 5.71	12.25 ± 1.23
	3 d	18.59 ± 3.34	21.33 ± 2.54	20.91 ± 3.43	16.59 ± 1.73	17.75 ± 2.89	197.18 ± 9.54	16.86 ± 2.82
	5 d	10.73 ± 1.58	13.48 ± 1.63	14.25 ± 2.16	9.62 ± 1.11	10.18 ± 2.05	173.83 ± 10.68	9.81 ± 1.35
	8 d	4.74 ± 1.08	6.75 ± 1.36	7.28 ± 1.64	3.42 ± 0.82	5.53 ± 0.56	105.55 ± 6.23	4.99 ± 0.86
	12 d	1.32 ± 0.45	2.62 ± 0.70	3.07 ± 1.50	0.79 ± 0.45	1.64 ± 0.70	41.16 ± 5.34	0.71± 0.16
	16 d	—	0.40 ± 0.23	0.86 ± 0.41	—	—	8.05 ± 1.76	—
	24 d	—	—	—	—	—	0.94 ± 0.32	—

**Table 3 molecules-28-04659-t003:** The main pharmacokinetic parameters after a single articular cavity administration of TPL-NS-Gel and TPL-NS at 10 mg/kg in RA rats (*n* = 3). The parameters were calculated using DAS 2.0 software.

Group	Non-Compartment Model			Two-Compartment Model		
	Parameters	Mean	SD	Parameters	Mean	SD
TPL-NS-Gel	AUC_(0–t)_ (μg/L·h)	3506.56	84.23	A	289.31	229.52
	AUC_(0–∞)_ (μg/L·h)	3858.09	345.14	B	299.05	173.35
	MRT_(0–t)_ (h)	247.11	18.01	α	0.01	0
	MRT_(0–∞)_ (h)	314.60	59.59	β	0.01	0
	t_1/2_z (h)	134.91	52.57	t_1/2_α (h)	69.32	4.08
	T_max_ (h)	192	0	t_1/2_β(h)	62.92	10.75
	C_max_ (μg/L)	12.26	2.66	AUC_(0–t)_ (μg/L·h)	3562.66	116.49
	CLz/F (L/h/kg)	2.61	0.22	AUC_(0–∞)_ (μg/L·h)	4092.18	420.39
	Vz/F (L/kg)	496.54	147.55	t_1/2_ Ka (h)	61.75	7.70
				CL/F (L/h/kg)	2.46	0.24
				K10 (1/h)	0.021	0.034
				K12 (1/h)	0.009	0.009
				K21 (1/h)	0.01	0
TPL-NS	AUC_(0–t)_ (μg/L·h)	2074.45	129.79	A	329.79	186.59
	AUC_(0–∞)_ (μg/L·h)	2123.78	114.74	B	283.17	173.27
	MRT_(0–t)_ (h)	106.58	4.20	α	0.025	0.005
	MRT_(0–∞)_ (h)	112.53	2.97	β	0.017	0.002
	t_1/2_z (h)	48.16	3.40	t_1/2_α (h)	28.40	5.12
	T_max_ (h)	64	13.86	t_1/2_β(h)	40.49	5.58
	C_max_ (μg/L)	17.13	3.02	AUC_(0–t)_ (μg/L·h)	2070.67	140.15
	CLz/F (L/h/kg)	4.72	0.26	AUC_(0–∞)_ (μg/L·h)	2147.80	156.62
	Vz/F (L/kg)	328.35	36.87	t_1/2_ Ka (h)	7.63	9.22
				CL/F (L/h/kg)	4.67	0.35
				K10 (1/h)	0.22	0.17
				K12 (1/h)	0	0
				K21 (1/h)	0.021	0.001

## Data Availability

The data presented in this study are available on request from the corresponding author.

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
