# Peer review of "Preparation, Properties and Therapeutic Effect of a TPL Nanoparticle Thermosensitive Gel for Intra-Articular Injection"

_molecules, 2023, doi:10.3390/molecules28124659_

Round 1
Reviewer 1 Report
The article "Production, properties and therapeutic effect of thermosensitive TPL nanoparticle gel for intra-articular injections" concerns research on the possibility of using a gel containing TPL nanoparticles for intra-articular injections in rhematoid arthritis. The properties of the TPL-NS-Gel gel were tested and its therapeutic effect was evaluated in a rat model of RA. The authors showed that after intra-articular injection, TPL-NS-Gel prolonged the release of the drug, reduced the concentration of the drug outside the joint tissue and improved the therapeutic effect in a rat model of RA. TPL-NS-Gel can be used as a new type of prolonged-release preparations for joint injections. The title of the article corresponds to its content, and the subject is within the profile of the journal. The authors indicate the purpose of the research and its results. The references are well-chosen, reflecting the work that has been carried out in the world in recent years.
1. The study answers the question: is the gel used a suitable agent that will gradually release the drug into the joint and is it beneficial for the patient?
2. This topic is interesting, but after some thought, I find that this article is more suited to a medical journal, as the space devoted to the study of the material is not very extensive.
3. The action of the gel is more revealing.
4. The captions under the drawings are too general, they do not help in understanding what is on them. in Fig 2C there are graphs that do not know what they refer to - you have to look for them in the text. Table 1 is hard to read, the numbers are merging.
5. References are relevant.
This manuscript appears to be useful for readers dealing with thermosensitive nanomaterials for biomedical applications. I think the manuscript is suitable for publication in Materials.
Reviewer 2 Report
The following suggestions must be incorporated:
1. Page 2 and Line 82. What was the low temperature?
2. Page 4 and Line 9. Mention the unit of chemical shift.
3. Figure 2 caption is incomplete. Figures 2A, B, C stand for what?
4. The image quality of Figure 2 B must be improved.
5. X and Y axis legends are missing in Figure 5.
6. Units are missing in Table 1.
7. Maintain the uniformity in expression of units throughout the manuscript, such as r·min-1 (Line 356) and r/min (Line 453)
8. Animal study Protocol approval No with date must be provided.
9. Drug release Kinetics is missing.
10. The current article seems to be a premature submission. A refinement is required.
11. Toxicity studies should be included.
12. Authors are suggested to incorporate a few more recent studies in the discussion section including 10.3390/pharmaceutics14091812
Minor editing of the English language is required.
Round 2
Reviewer 2 Report
The authors have revised their manuscript by responding reviewer's queries at a level of satisfaction.